# Portable molecular diagnostic platform for rapid point-of-care detection of mpox and other diseases

Matthew L. Cavuto [1,2,10], Kenny Malpartida-Cardenas [1,2,10], Ivana Pennisi[1,2], Marcus J. Pond[3], Sohail Mirza[1,2], Nicolas Moser[2,4], Mark Comer[2], Isobel Stokes [5], Lucy Eke[5], Sian Lant [5], Katarzyna M. Szostak-Lipowicz [2], Luca Miglietta [1], Oliver W. Stringer[1,2], Katerina-Theresa Mantikas[2,4], Rebecca P. Sumner [5], Frances Bolt[1,2], Shiranee Sriskandan [1,6], Alison Holmes [1,7,8], Pantelis Georgiou[2,4], David O. Ulaeto [9], Carlos Maluquer de Motes [5] & Jesus Rodriguez-Manzano [1,2] ✉

The World Health Organization's designation of mpox as a public health emergency of international concern in August 2024 underscores the urgent need for effective diagnostic solutions to combat this escalating threat. The rapid global spread of clade II mpox, coupled with the sustained human-to-human transmission of the more virulent clade I mpox in the Democratic Republic of Congo, highlights a critical gap in point-of-care diagnostics for this emergent disease. In response, we developed Dragonfly, a portable molecular diagnostic platform for point-of-care use that integrates power-free nucleic acid extraction (<5 minutes) with lyophilised colourimetric LAMP chemistry. The platform demonstrated an analytical limit-of-detection of 100 genome copies per reaction for monkeypox virus, effectively distinguishing it from other orthopoxviruses, herpes simplex virus, and varicella-zoster virus. Clinical validation on 164 samples, including 51 mpox-positive cases, yielded 96.1% sensitivity and 100% specificity for orthopoxviruses, and 94.1% sensitivity and 100% specificity for monkeypox virus. Here, we present a rapid, accessible, and robust point-of-care diagnostic solution for mpox, suitable for both low- and high-resource settings, addressing the global resurgence of orthopoxviruses in the context of declining smallpox immunity.

In 2022, as the SARS-CoV-2 pandemic waned, a global mpox outbreak emerged, exhibiting clear epidemiological and clinical differences from historical mpox cases, which were typically associated with zoonotic acquisition and limited human transmission[1,2]. For the first time, sustained human-to-human transmission was observed, primarily through sexual contact, affecting both non-endemic and endemic regions[1,3–6]. This unprecedented outbreak was declared a public health emergency of international concern (PHEIC) by the World Health Organization (WHO) in July 2022[3], rapidly spreading across 121 countries and resulting in 99,518 confirmed cases and 207 deaths[4,7]. The PHEIC was declared over in May 2023 after a sustained decline in global cases. However, in 2024, the number of cases surged again, with more than 20,000 new cases and over 600 deaths reported[8,9]. Notably, a more virulent strain, clade Ib, was identified in several countries, causing more severe disease with a higher mortality rate compared to clade II, which was predominantly responsible for the 2022 outbreak. Considering the upsurge of mpox, the growing number of countries being affected in Africa, and the potential to

spread further outside the continent, the WHO once again declared a PHEIC in August 2024[8,10].

An initial and sustained lack of detection is thought to have played a significant role in the global dissemination of clade II mpox[11]. The fast-paced global spread of clade II mpox and its evolving epidemiology from traditionally zoonotic to human-adapted transmission[12] appears to be mirrored by the new clade Ib, which is spreading at alarming rates within the Democratic Republic of the Congo (DRC) and neighbouring countries[13]. This change underscores the critical importance of early case detection and population surveillance to support effective pandemic preparedness and outbreak management[13–15]. Post hoc investigations into responses to similarly novel and evolving infectious outbreaks, such as HIV[16,17], influenza virus[18,25], Ebola[19], and SARS-CoV-2[20], have consistently placed significant emphasis on the role of diagnostics in successful control, and highlighted the need for solutions suitable for use at the point-of-care (POC) and in low-resource settings[21].

The WHO 'Strategic framework for enhancing prevention and control of mpox 2024–2027' aims to end human-to-human transmission of mpox to remove the public health and pandemic threat that mpox presents[7]. The framework requires the development of diagnostic assays for use in decentralised sites or the POC. The WHO Target Product Profiles for mpox diagnosis[4,7,22] states the importance of differential tests that distinguish multiple diseases, particularly in the prodrome phase when mpox clinical presentation may overlap with other conditions. Distinguishing mpox from similar skin conditions, including chickenpox and herpes simplex, during mpox's eruptive phase is also critical in identifying cases, informing infection management, and reducing onward transmission[22,23]. Preliminary mpox diagnosis is often made clinically, with confirmation by molecular nucleic acid amplification tests (NAATs) or viral culture[24,25], which severely limits diagnostic solutions in remote and resource-limited settings where the endemic disease burden is greatest[26,27].

Protein-based immunoassays, such as lateral flow tests (LFTs), provide a rapid, portable, and cost-effective solution for POC diagnostics and are valuable for estimating population-level infection rates[28,29]. However, retrospective investigations into diagnostic technologies, including those used during the COVID-19 pandemic, have consistently highlighted sensitivity limitations with LFTs[30–32]. At present, there is no LFT available with the necessary sensitivity for mpox detection. Consequently, NAATs, such as quantitative PCR (qPCR), have remained the gold standard for mpox diagnosis due to their excellent specificity and sensitivity[33]. Despite these advantages, however, qPCR presents notable limitations, predominantly in POC and near-patient use[34].

A selection from the limited number of commercially available near-POC and true-POC molecular mpox diagnostics are listed in Supplementary Table 1, as identified by FIND[35]. These examples include PCR panels targeting MPXV, OPXV, or other pathogens (such as VZV, or HSV) that are then paired with expensive automated systems such as Cepheid GeneXpert®, QIAGEN QIAStat-Dx Analyzer, Ustar EasyNAT, or Wondfo U-Card Dx. While the majority of these platforms provide a simplified sample-to-result workflow with minimal hands-on time, the incorporation of sophisticated electro-mechanical automation leads to bulky and expensive devices, typically confining them to well-funded, controlled, and centralised laboratory settings. On the other hand, strictly laboratory-based assays compatible with even larger conventional instruments have further drawbacks, including the need for a cold chain and substantial manual input from skilled technicians. All these factors limit the portability and accessibility of such diagnostic tools, hindering their deployment in emergencies, both at the POC and in low-resource settings[36].

Loop-mediated isothermal amplification (LAMP) based assays can offer an important alternative to qPCR[37–40]. Notably, LAMP is a NAAT

approach that operates under isothermal conditions, with the amplification reaction occurring at a constant temperature (typically 60 to 65 °C), eliminating the need for a thermocycler and making LAMP a more accessible, cost-effective, and portable option for POC use. Nonetheless, simplifying LAMP-based NAATs for POC use, without compromising accuracy and sensitivity, remains a technological challenge, with sample processing in particular presenting a significant bottleneck[41–43]. Sensitivity has been shown to depend heavily on the quality, purity, and concentration of nucleic acids in the processed sample[43,44], requiring complex sample preparation protocols to ensure successful amplification. While recent efforts have examined extraction-free, or "direct", NAATs to reduce cost and complexity, these have been shown to sacrifice sensitivity through increased amplification inhibition and sample dilution[45–47].

To address existing diagnostic gaps, we have developed and validated Dragonfly, a portable, molecular sample-to-result POC diagnostic platform designed for the rapid multi-pathogen detection and differentiation of skin-tropic viruses. Our platform incorporates a simple power-free nucleic acid extraction and purification method based on magnetic beads[48,49], which is coupled with lyophilised colourimetric LAMP technology[50]. This approach significantly reduces time-to-result (under 40 minutes) and eliminates the need for cold-chain storage. It also minimises hands-on time and the requirement for complex instrumentation. Unlike alternative diagnostic solutions, Dragonfly only requires an isothermal heat block, eliminating the need for bulky and expensive instruments, and providing a truly POC format that is convenient and accessible.

Here, we present a sample-to-result platform for the simultaneous detection and differentiation of orthopoxvirus (OPXV) genus, mpox (clades I and II), varicella-zoster virus (VZV), herpes simplex virus type 1 (HSV-1), and herpes simplex virus type 2 (HSV-2) in a multiplex skin panel, enhancing diagnosis and infection management. The platform's performance was validated using 164 clinical samples, including 51 mpox clade II positive samples, and was benchmarked against a gold-standard extracted qPCR workflow. Our platform demonstrated high analytical performance, achieving 96.1% sensitivity and 100% specificity for OPXV detection, and 94.1% sensitivity and 100% specificity for MPXV detection.

## Results
### Platform overview
The developed platform is comprised of three core components: a single-use sample extraction kit, a lyophilised colourimetric LAMP-based panel, and a low-cost isothermal heat block, as shown in Fig. 1a. To support a POC workflow, the platform also includes a sample collection kit (containing a swab and inactivating medium, COPAN eNAT®), a reusable fixed-volume pipette with disposable tips, and an optional tablet with cloud-connected companion software for result logging and centralised data management. The Dragonfly workflow includes power-free extraction of nucleic acids in under 5 minutes, followed by colour-based molecular detection in less than 35 minutes, culminating in a visual equipment-free result read-out. A high-level overview of the complete workflow is shown in Fig. 1b, c, with the core nucleic acid extraction technology, SmartLid, highlighted in Fig. 2, and photographs of select process steps, from sample input to result readout, provided in Fig. 3a–f.

### Platform components
Previously developed by our group[48,49] and as depicted in Fig. 2, the SmartLid technology utilises a magnetic lid to capture and transfer superparamagnetic nanoparticles (or magnetic beads) and attached DNA/RNA through three simple steps (lysis-binding, washing, and elution), enabling efficient and power-free nucleic acid extraction without centrifugation or manual pipetting. All required buffers are pre-aliquoted in colour-coded tubes (utilising a sequential traffic-light

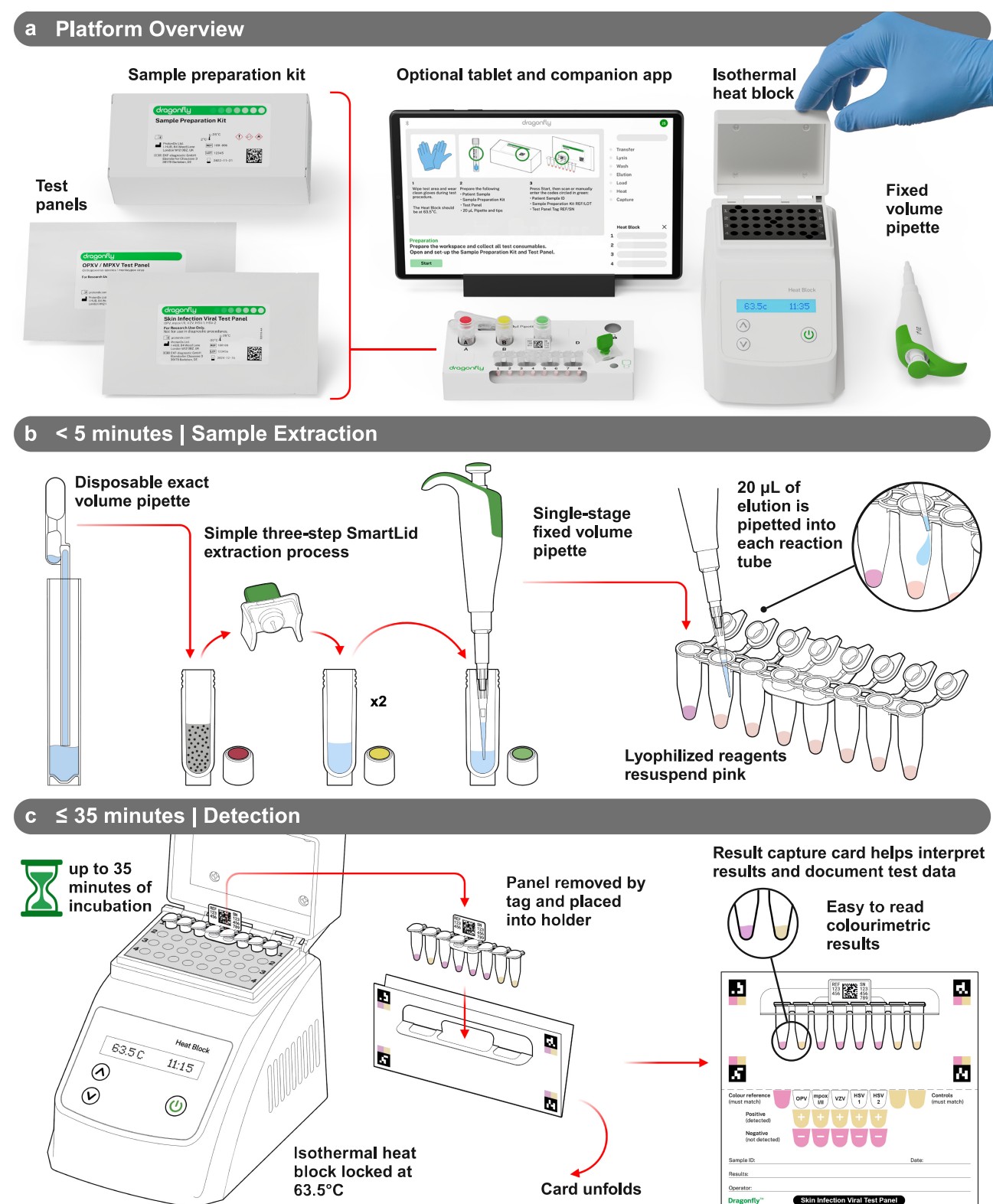

**Fig. 1 | The Dragonfly platform. a** Overview of the platform, including consumable components, optional tablet and companion app, as well as deployed disposable mobile workstation, and low-cost isothermal heat block with reusable fixed-volume pipette. **b** High-level overview of rapid nucleic acid extraction process and test panel loading for one sample. **c** High-level overview of test panel incubation and colourimetric result interpretation. Created in BioRender. Cavuto, M. (2025) https://BioRender.com/d45n457.

system of red, yellow, and green) and packaged in a cardboard tray (150 × 70 × 50 mm) along with disposable exact-volume pipettes for sample input and a SmartLid for magnetic bead manipulation. In Fig. 3c, we show that the fully recyclable carboard packaging also

functions as a workstation, facilitating the POC workflow with clear diagrammatic labelling, a space for housing the sample tube, and receptacles for holding the lyophilised test panel during reagent resuspension.

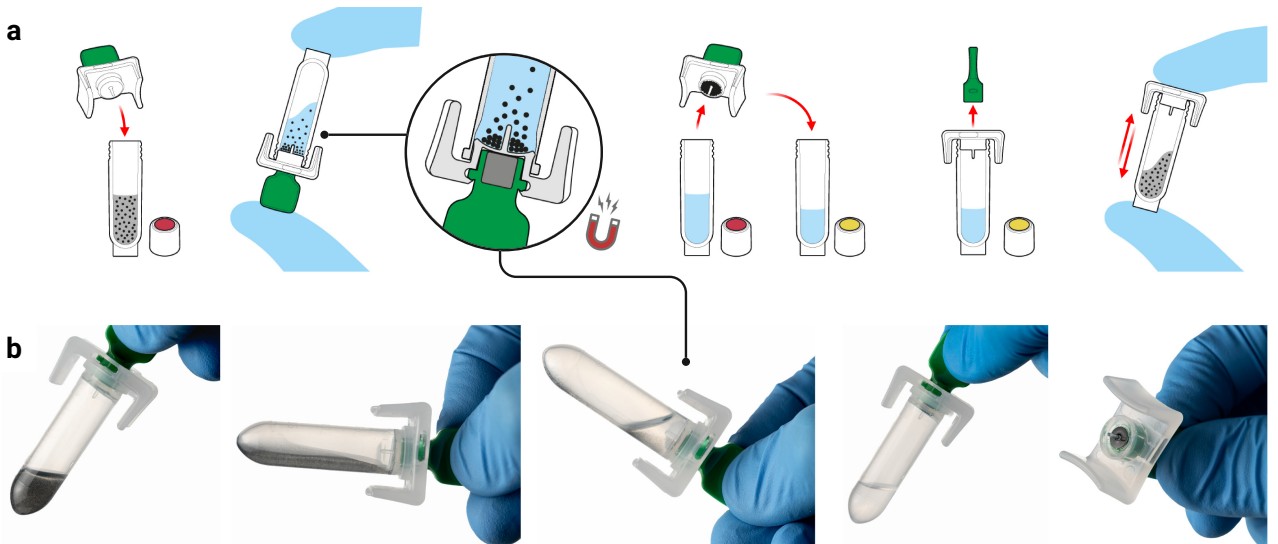

**Fig. 2 | The SmartLid nucleic acid extraction technology. a** Graphical illustration of SmartLid usage to transfer magnetic beads from one tube to another with a removable magnet. **b** Close-up images showing the rapid magnetic bead collection process, with beads visible on the underside of the SmartLid after collection. Created in BioRender. Cavuto, M. (2025) https://BioRender.com/d45n457.

For the test panel, a lyophilised colourimetric LAMP chemistry was developed to yield long term room temperature storage, visual result readout, and isothermal incubation[50]. This enables visual identification of positive reactions based on proton production (and subsequent pH drop), which occurs with nucleotide incorporation during DNA polymerase activity. Furthermore, this colour transition, from pink (at high pH) to yellow (at low pH), is compatible with multiple colour vision deficiency (CVD) friendly colour schemes, ensuring maximum usability regardless of user profile[51]. An off-the-shelf eight-tube PCR strip (4titude®), with individually flip-capped 0.2 mL tubes, was selected to house each pre-dispensed and lyophilised reaction master mix. To enhance traceability, polycarbonate tags, attached in the middle of each tube strip, provide space for data-matrix and alpha-numeric labelling. Combined, these three aspects create a platform that minimises reliance on equipment, requiring only a low-cost, portable, and user-friendly isothermal heat block (160 × 110 × 130 mm, <1 kg). This heat block can be powered by mains electricity, a standard 12-volt supply, batteries, or solar panels, drawing less than 20 W continuously once at the correct temperature.

### Cloud-connected companion application

To augment the manual workflow, an optional companion Android application was developed with the goal of improving the user experience and enabling wireless cloud connectivity to integrate with healthcare databases (Fig. 3g–j). Utilising the onboard device camera, the application further enhances traceability by scanning data matrices on all consumables at the start of the process. A virtual step-by-step workflow was implemented to guide new users through the process, with progress tracked and displayed, and animated clickable timers provided throughout the workflow to time each step (e.g., buffer shaking). Test panel incubation is also monitored and timed, with assistance provided to remind the user of the incubation status of each sample and their respective location in the heat block. After incubation, result interpretation and recording were augmented through an image capture process, in which a cropped and enlarged view of the eight tubes is presented along with buttons to select observed colours. Depending on the specific panel scanned in at the beginning of the process, results are then interpreted and displayed. All results, along with captured test

panel images, are automatically stored and summarised in an AWS cloud-accessible dashboard for export, review, and traceability. Further details of the Dragonfly application are provided in Supplementary Methods.

### Test Panel Design

Our Skin Infection Viral Test Panel was designed to target orthopoxvirus genus (OPXV), monkeypox virus (MPXV), varicella-zoster virus (VZV), herpes simplex virus type 1 (HSV-1), and herpes simplex virus type 2 (HSV-2). VZV, HSV-1 and HSV-2 can cause skin rashes and lesions potentially confounding mpox diagnosis. In addition, zoonotic OPXVs such as cowpox and borealpox have been associated with fatal human infections in Europe and America[52]. Therefore, our panel aims to facilitate the accurate diagnosis of mpox cases while capturing the emergence of zoonotic OPXV species. The panel layout is shown in Fig. 1c and includes the following targets, from left to right: colour reference control, OPXV, MPXV, VZV, HSV-1, HSV-2, extraction control, and internal control.

Two LAMP assays targeting two distinct genomic regions were included per lyophilised reaction mix for OPXV, MPXV, and VZV, with one assay each for HSV-1 and HSV-2. For detection of OPXV, we selected *E9L* (viral DNA polymerase gene, a conserved segment across all Eurasian OPXVs) and *F13L* (encoding a conserved protein essential for viral maturation and release from infected cells[53]), with *E9L* also used as the target in the US Centers for Disease Control and Prevention (CDC) qPCR assay[54]. For MPXV, assays targeting conserved intraspecies regions of *G2R* (a viral tumour necrosis factor receptor) and *A9L* (a VACV orthologue encoding a morphogenesis factor) were designed to cover both clade I and II without cross-reactivity with other OPXVs. *G2R* is the target currently used by the CDC qPCR assay for MPXV detection[55]. For VZV, an assay targeting the *ORF28* gene was designed, along with an additional assay targeting the *ORF62* gene, as described by Okamoto et al.[56]. For HSV-1, an assay from Kaneki et al.[57] targeting the *UL1* gene was used, and for HSV-2, we developed an assay targeting the *US4* gene. LAMP assay sequences are provided in Supplementary Table 2.

The panel includes three additional control reactions to ensure reliable results. First, as Dragonfly utilises a pH-based colourimetric indicator, variations in sample and extracted elution pH were observed to affect the starting resuspended (negative) reaction colour. To

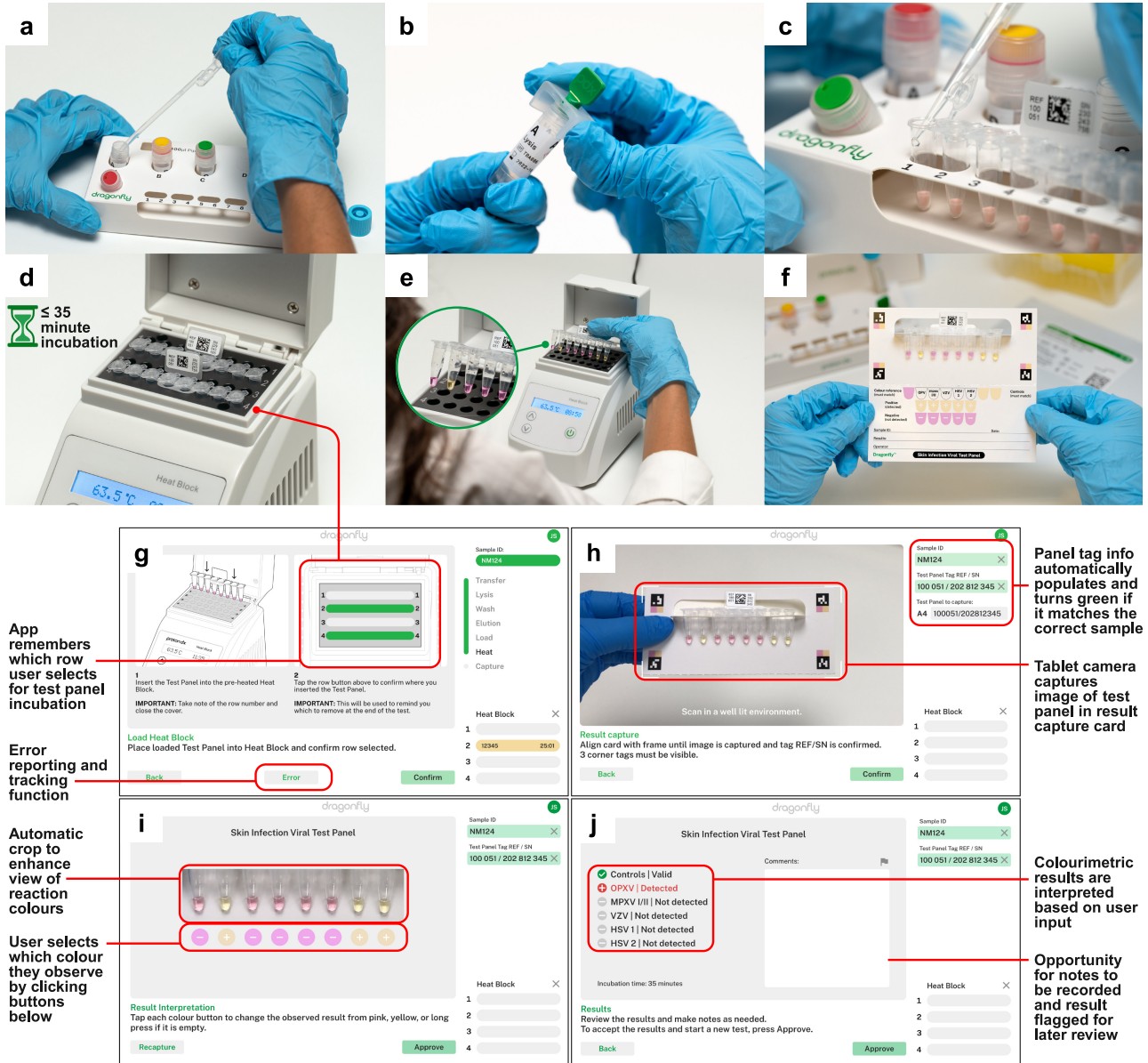

**Fig. 3 | The Dragonfly sample-to-result workflow. a** Sample input using a disposable exact-volume pipette. **b** Insertion of the SmartLid to initiate the rapid power-free extraction process. **c** Close-up of extracted nucleic acids being loaded into the open test panel using a 20 μL fixed-volume pipette. **d** Simultaneous incubation of two test panels, with a capacity of up to four panels per low-cost, portable isothermal heat block. **e** Fully incubated colourimetric test panel being removed from the heat block. The attached panel tag allows for easy removal, reduces the risk of accidental tube opening, and verifies correct panel selection through a comparison with the scanned tag at the beginning of the process. **f** Example of a test panel result indicating a positive (yellow) reaction for OPXV, along with valid test controls. Screenshots from the companion application, showing the test panel loading screen (**g**), incubated test panel result capture screen (**h**), result confirmation screen (**i**), and result interpretation and recording screen (**j**). Created in BioRender. Cavuto, M. (2025) https://BioRender.com/d45n457.

account for this, a "colour reference control" was added (which excludes amplification enzymes), ensuring that one tube always remains pink, and thus represents a pH-adjusted negative reference colour. Next, an "internal control" reaction mix was added, which contains the DNA template specific to its assay, ensuring amplification under ideal operating conditions. This control acts as a confirmatory reaction to indicate that the test panel is in good working order (e.g., not damaged due to improper storage, incubated at the wrong temperature, or excessively inhibited). Finally, a "human extraction control" was included, targeting the human housekeeping *beta-actin* gene[58], providing a confirmation that the extraction process was performed correctly and that the sample source was adequately swabbed. A valid test result, whether positive or negative, requires that the three

described control reactions are pink (negative), yellow (positive), and yellow (positive), respectively.

## Workflow

The nucleic acid extraction and molecular detection protocols were optimised to achieve a balance between workflow complexity, time, and performance: Inactivated swab eluent (400 μL) is transferred from the sample collection tube into Tube A (containing lysis-binding buffer and magnetic beads) using a disposable exact volume pipette (Fig. 3a). The included SmartLid is then used to transfer magnetic beads and their attached nucleic acids through a series of three sample extraction steps (A: lysis-binding, B: wash, and C: elution) (Fig. 3b). At each step, resuspended magnetic beads are mixed with the buffers for

30 seconds through manual shaking without the magnet, followed by insertion of the magnet to collect the magnetic beads, and a 30 second drying step to evaporate any remaining solvents prior to elution. Once nucleic acids are eluted into Tube C, the magnetic beads are removed with the SmartLid. Next, a 20 μL fixed volume pipette (using a disposable tip) is used to transfer elution into each tube of the test panel, resuspending the lyophilised reagents (Fig. 3c). Unlike laboratory micro-pipettes, the included fixed-volume pipettes were modified to remove the secondary 'blow-out' stage, simplifying the pipetting process without affecting test sensitivity or repeatability. Once all caps are firmly closed, test panels are placed into one of the four heat block rows (Fig. 3d) and incubated for 35 minutes at 63.5 °C (heater locked at temperature to reduce potential user error). After incubation, test panels are placed into a result capture card, where developed colours can be compared to a key directly below each tube, indicating the result: pink for negative, yellow for positive. (Fig. 3e, f) The entire process, from sample-to-result, was optimised to take <40 minutes per sample, with each subsequent sample able to be extracted while previous samples are being incubated, yielding a continuous throughput of greater than 12 samples per hour per user.

### Analytical sensitivity and specificity of LAMP assays

Analytical sensitivity was evaluated using serial dilutions of synthetic DNA (one for each target), and the obtained standard curves had a correlation ($R^2$) of 80 to 99% (shown in Supplementary Fig. 1). Assay optimisation results are provided in Supplementary Data, assays sequences in Supplementary Table 2, and synthetic DNA sequences in Supplementary Table 4. Time-to-positive (TTP) values across all tested concentrations ranging from $10^1$ to $10^7$ copies per reaction were within 15 minutes except for HSV-2, which required 25 min. All assays had an overall limit of detection (LOD) from 10 to 500 copies per reaction as shown in Fig. 4b and Supplementary Table 3. Analytical specificity was experimentally assessed using extracted nucleic acids from commercially available viral particles (Vircell, MBTC032-R, Zeptometrix CATALOG# NATHSV-6L and Zeptometrix CATALOG# NATVZV-STQ). As shown in Fig. 4c, the LAMP assays only amplified their specific targets.

### Sample-to-result evaluation with viral particles

Commercially available MPXV and HSV-1/HSV-2/VZV viral particles (Vircell, MBTC032-R and MBTC016) were spiked into eNAT® (COPAN) inactivation buffer at various concentrations, as shown in Supplementary Table 5. Extractions were performed using Dragonfly Sample Preparation Kits, with eluted nucleic acids used to resuspend our Skin Infection Viral Test Panels, which were then incubated for 35 minutes. The LOD for OPXV and VZV was determined to be 50 copies per reaction, equivalent to $1.25 \times 10^3$ copies per mL, assuming 100% extraction efficiency. The LODs for MPXV and HSV-1 were determined to be 100 copies per reaction, and 143 copies per reaction for HSV-2. Assuming 100% extraction efficiency, this corresponds to $2.50 \times 10^3$ and $3.58 \times 10^3$ copies per mL for MPXV/HSV-1 and HSV-2, respectively. Examples of these results are shown in Fig. 4e. Lastly, specificity of the panel was further confirmed using cowpox (CPXV) and vaccinia (VACV) viral particles at concentrations of $5 \times 10^5$ PFU per mL, where appropriately only the OPXV reactions turned yellow.

### Virucidal activity of eNAT® buffer against VACV and HSV-1

As shown in Fig. 4a, after 2 minutes of incubation at room temperature, eNAT® buffer demonstrated significant virucidal activity against VACV, achieving a reduction in viral titters by ≥8.0 $\log_{10}$ TCID50/mL. This rapid inactivation indicates that the buffer is highly effective at neutralising the virus under the tested conditions, suggesting it is suitable for safe sample handling and processing. Additionally, the eNAT® buffer exhibited potent virucidal activity against HSV-1. Following a slightly longer incubation period of 5 minutes at room temperature, viral titters were reduced by ≥7 $\log_{10}$ TCID50/mL, further confirming

the buffer's efficacy across multiple virus families. See Supplementary Fig. 2 for a summary of the results.

### Testing of clinical skin lesion swabs

A total of 164 surplus extracts from skin lesion swabs, which had been submitted to North West London Pathology (NWLP) for clinical testing, were utilised to assess whether the described viral DNA targets could be detected using the Dragonfly platform. These results were compared with identifications obtained via gold-standard automated nucleic acid extraction and real-time multiplex qPCR detection. The samples, collected in Roche COBAS PCR media (P/N: 07958030190) included 51 mpox clade II positive samples and 40 samples positive for one or more herpes simplex virus. Our platform demonstrated high analytical performance, with 96.1% (95% CI of 86.5% to 99.5%) sensitivity and 100% (95%CI of 96.8% to 100%) specificity for OPXV, and 94.1% (95%CI of 83.8% to 98.8%) sensitivity and 100% (95%CI of 96.8% to 100%) specificity for MPXV. Distributions of the $C_t$ values for positive samples are shown in Fig. 4d. There were two false negatives for OPXV and MPXV, both of which had qPCR $C_t$ values above 33 (34.37/33.61 and 35.96/34.97, for OPXV and MPXV respectively) and one false negative exclusively for MPXV with a qPCR $C_t$ value of 28.65. The complete dataset, with results for all targets and clinical samples, is provided as Supplementary Data.

The swabs included 10 VZV, 20 HSV-1, and 10 HSV-2 infections, as identified by qPCR, representing 24.4% (40/164) of the samples analysed. This included 4/164 (6.3%) co-infections with mpox (three with HSV-1, and one with HSV-2). The Dragonfly platform detected 9/10 VZV, 18/20 HSV-1, and 7/10 HSV-2 cases (85% of the positive samples), including three out of the four co-infections. Confusion matrices of diagnostic performance are provided in Supplementary Fig. 3. All samples yielded a positive result for the human extraction control reaction (targeting the beta-actin gene), confirming the quality of the extraction and the adequacy of sample collection, limiting the potential for false negatives due to user error or insufficient sampling. To verify the results of the human extraction control reaction, a TaqMan assay targeting the RNase P gene was used for the qPCR comparator, with a distribution of $C_t$ values shown in Supplementary Fig. 4.

## Discussion

Mpox, an infection caused by MPXV, is historically associated with localised outbreaks in endemic countries of central, west, and east Africa, with limited human-to-human transmission. In 2022, a global mpox clade II outbreak was declared a PHEIC by the WHO. After the emergency was lifted in May 2023 due to a decline in cases, a resurgence occurred in 2024, driven by a more virulent strain, clade Ib. In response to this upsurge, the WHO reinstated the PHEIC in August 2024. The rapid global spread and altered epidemiology of these outbreaks has highlighted the importance of early case detection and population surveillance to support effective pandemic preparedness and management[8,12].

In this study, we developed and validated a truly-POC molecular diagnostic system, Dragonfly, for the accurate simultaneous detection of mpox and other associated skin tropic viral diseases to support the differential diagnosis of mpox from OPXV genus, HSV-1, HSV-2 and VZV, all of which present with similar skin lesions. Our platform combines simple power-free nucleic acid purification[48,49] with lyophilised colourimetric LAMP technology[50] to achieve PCR grade analytical performance away from centralised laboratories, and without specialised laboratory equipment, a cold chain, or skilled laboratory personnel. Dragonfly significantly reduces the reliance on bulky and expensive equipment, forgoing complex optical fluorescent systems and thermocyclers and enabling use in remote and resource-limited settings. Our Skin Infection Viral Test Panel was validated using 164 clinical skin lesion samples, including 51 mpox clade II positive samples, 10 VZV, 20 HSV-1, and 10 HSV-2 positive samples. The results were

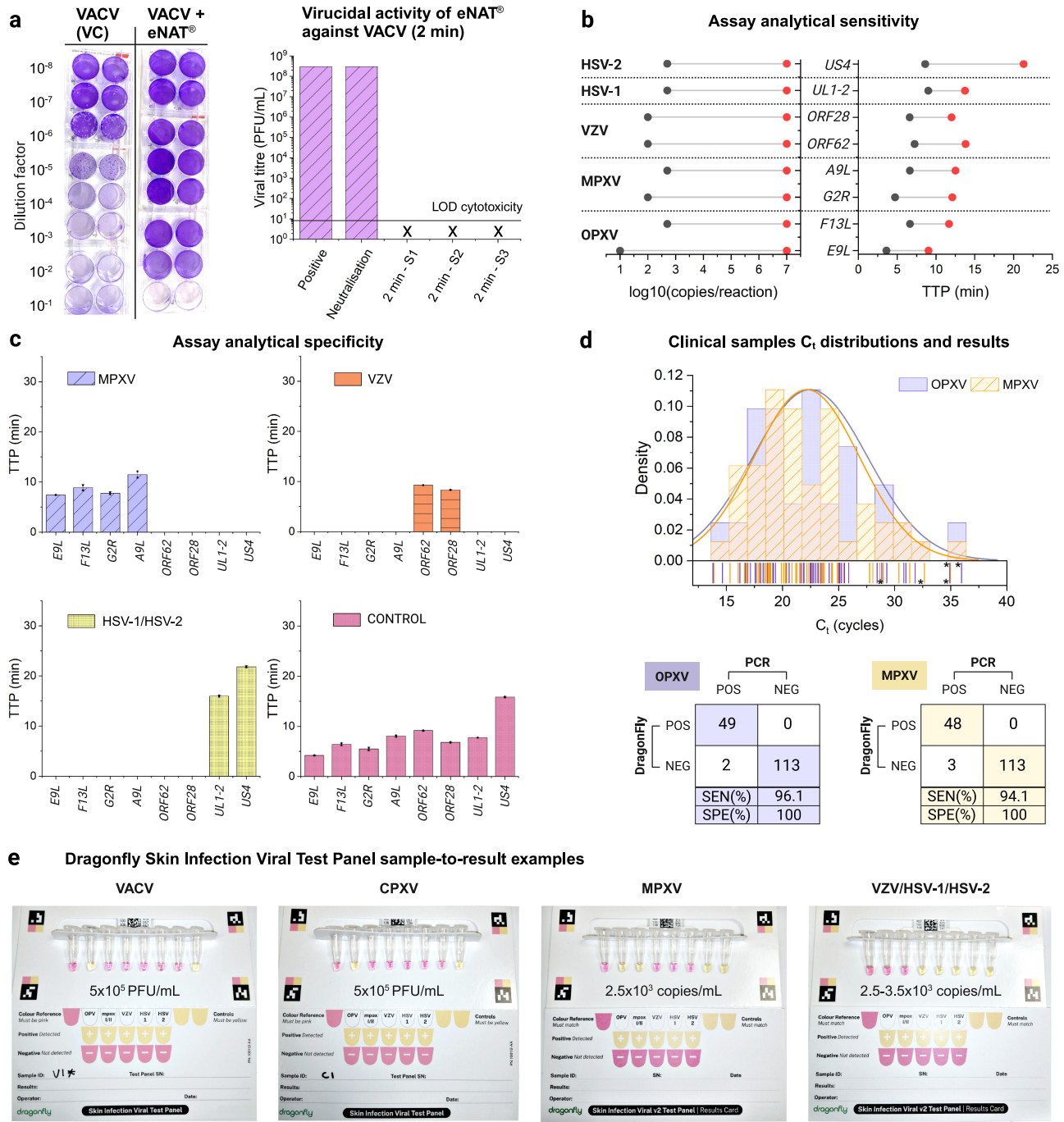

**Fig. 4 | Validation of the Dragonfly Skin Infection Viral Test Panel. a** Evaluation of virucidal activity of eNAT® against vaccinia virus (VACV): Plaque assays were performed using confluent monolayers of BSC40 cells. The bar plot shows the virucidal activity after a 2-minute exposure, showing 8-log reduction, alongside images of crystal violet-stained plaque assays (1% crystal violet, 70% ethanol) after 30 minutes. **b** Analytical sensitivity of LAMP assays: range of detected concentrations of synthetic DNA (log10 of copies per reaction) and corresponding TTP values in minutes. **c** Analytical specificity of LAMP assays: Using extracted nucleic acids from various commercially available viral particles (MXPV, HSV-1/HSV-2, and VZV), and a control sample consisting of synthetic DNA at $5 \times 10^3$ copies per reaction. The bar plot displays mean TTP values and data points (in minutes) with error bars representing the standard deviation (SD). **d** qPCR $C_t$ values distribution: A histogram illustrating the distribution of qPCR $C_t$ values obtained from OPXV and MPXV-positive clinical samples (purple dots for OPXV and yellow dashed lines for MPXV). Confusion matrices of the diagnostic performance are included below. **e** Sample-to-result demonstration: Images showing the results for vaccinia (VACV), cowpox (CPXV), MPXV, and combined HSV-1/HSV-2/VZV viral particles, with spiking concentrations indicated below each set of reactions. VACV and CPXV viral particles, both spiked at $5 \times 10^5$ PFU/mL, demonstrate the specificity of the OPXV assay, showing no cross-reactivity with other target assays. Similarly, spiking MPXV ($2.5 \times 10^3$ copies/mL) and combined HSV-1/HSV-2/VZV viral particles ($2.5 \times 10^3$ copies/mL for VZV and HSV-1, $3.5 \times 10^3$ copies/mL for HSV-2) showed no cross-reactivity. Created in BioRender. Cavuto, M. (2025) https://BioRender.com/d45n457.

compared to gold-standard automated extraction and TaqMan-based qPCR assays, demonstrating high sensitivity and specificity.

The study was limited, however, by the low number of available confirmed HSV-1, HSV-2 and VZV samples. This was a result of the samples being collected at the height of the mpox outbreak and thus during a temporary decline in HSV/VZV testing. A wider validation for these targets should be examined as part of future studies. Additionally, our system was designed to extract samples from eNAT®

inactivation buffer, while all collected samples for this study were stored in Roche COBAS PCR media. Accordingly, samples were diluted 1:2 in eNAT® inactivation buffer to remain consistent with the intended use of the platform. Should samples be collected from patients and directly stored in eNAT® in future studies, the LOD would be expected to improve proportionally to the dilution factor above. Furthermore, all clinical mpox positive samples examined were clade II, and collected from the same outbreak. Further work will be required to understand the analytical performance on a broader strain collection. While preliminary in-silico analysis and experimental testing with synthetic DNA confirmed that our assays cover clade I, this will nonetheless require replication in a clinical setting. Finally, although the panel was designed for this study to include mpox clades I and II in the same reaction (i.e. to not differentiate clades I and II), it would be a simple matter to reconfigure the panel to achieve differentiation (for example, by instead combining the HSV-1 and HSV-2 assays into one reaction) or include additional targets such as molluscum contagiosum which can cause lesions similar to mpox, should that be deemed necessary.

The infectious disease landscape is dynamic, marked by constant changes in pathogen behaviour, emerging threats, and shifts in environmental and socioeconomic factors requiring diagnostic platforms to adapt[59]. Recent data describes a growing clade I mpox outbreak that is spreading in a new epidemiologic pattern analogous to that of the global clade II outbreak. Contrary to enclosed and complex microfluidic cartridges used in many other POC diagnostic systems[60], Dragonfly can responsively adapt to emerging needs. It has an easy to manufacture design, with extraction and amplification reagents that are rapidly swappable and customisable within generic off-the-shelf packaging solutions. This creates an adaptable system that can be repurposed for different sample types (e.g., blood, stool, etc.), targets (e.g., viral, bacterial, etc.), and applications in response to emerging requirements. For example, the Skin Infection Viral Test Panel described in this study was recently adapted into a dual-sample two-pathogen panel for higher throughput processing and reduced costs (Fig. 5). Alongside an updated companion application suitable for personal mobile phone use (Fig. 5a and Supplementary Methods), a dual-sample vortex tool was developed (Fig. 5b–d) to augment the already rapid diagnostic workflow. An associated training video for this version can be found at www.youtube.com/watch?v=Oz1kvwJyNzQ.

The robustness of a diagnostic test is a key enabler for its accessibility and utility in resource-limited settings[61,62]. Traditional molecular tests often require stringent transport and storage conditions, such as refrigeration, to maintain the stability of their many temperature-sensitive reagent components, creating logistical challenges[63]. In contrast, our platform utilises patent pending lyophilised colourimetric LAMP technology[50], which enables room temperature shipping and storage, ensuring the adaptability of the diagnostic platform to diverse environmental conditions, including where cold chain logistics are not available. Preliminary platform robustness details are provided in Supplementary Methods and Supplementary Table 7. Moreover, the compact and lightweight nature of the Dragonfly platform, weighing less than 1 kg (including the isothermal heat block) and fitting in a backpack (see Supplementary Fig. 5), facilitates on-the-go testing and allows healthcare professionals to reach remote or underserved areas efficiently.

In the years since the publication of the original ASSURED (affordable, sensitive, specific, user-friendly, rapid, equipment-free, deliverable) criteria for POC diagnostics in resource limited settings, connectivity and technological integration have become more widespread and provide opportunities for improved real-time diagnosis, surveillance, and monitoring[64]. A user-friendly companion app was therefore developed with cloud-based data storage and a dashboard for result visualisation and real-time data integration. Such connectivity solutions not only increase quality assurance for POC tests,

but also allow for centralised and real-time decision-making, even across tiered laboratory systems during outbreak investigations and global health emergencies.

Finally, cost-effective diagnostic platforms can be game-changers in the landscape of molecular sample-to-result systems. The primary equipment required by the majority of portable molecular platforms can cost tens of thousands of pounds. This high price point is a significant barrier for accessibility and widespread adoption, limiting the reach of molecular platforms across healthcare sectors and geographical locations. In contrast, the simple isothermal heater, the most expensive component of our system, is manufacturable for under £100, enhancing accessibility and economic viability on a large scale.

As emphasised by the recent WHO Strategic framework for enhancing prevention and control of mpox (2024–2027)[7], there is a need for POC solutions that can support early mpox detection and disease surveillance, especially in resource-limited settings where access to diagnostics is more restricted[13,28]. Although there has been an increase in the adoption of isothermal chemistries such as LAMP[39,65,66] or RPA in combination with CRISPR[67], such approaches are still primarily laboratory-based, requiring trained personnel, sample transport to centralised facilities, expensive equipment, and cold chain logistics. The Dragonfly platform's accuracy, portability, and rapid multi-pathogen sample-to-result capability make it a versatile tool with significant potential to contribute to global efforts in combating both emerging and endemic infectious diseases, particularly in low-resource environments.

## Methods

### LAMP assay design

LAMP assays were designed using Primer Explorer version 5.0 (http://primerexplorer.jp/lampv5e/index.html), based on sequences retrieved from NCBI GenBank database (https://www.ncbi.nlm.nih.gov/genbank/). Sequences were aligned using the MUSCLE algorithm[68] in Geneious 2023.1.2, with several primer sets being designed and evaluated for each target gene. Primer sets were assessed based on melting temperature ($T_m$), GC content, and absence of secondary structures, which was verified through the built-in DNA Fold tool in Geneious. Primer candidates were tested experimentally and optimised to perform at 63.5 °C, which is the operating temperature of the Dragonfly platform. The optimal primer sets were selected based on their performance in LAMP reactions, considering factors such as amplification efficiency (TTP) and specificity (absence of amplification in non-template controls). Data is provided as Supplementary Data, and final primer sequences in Supplementary Table 2. All primers were purchased from Integrated DNA Technologies (IDT) and rehydrated in nuclease-free water at 400 μM. The LAMP assay specific to the human house-keeping gene *beta-actin*[58], LAMP-ACTB, and LAMP assays specific to HSV-1 and VZV (*ORF62*), were previously described[56,57]. Reaction end-point time was determined after evaluating the analytical sensitivity of the assays and chosen to be the TTP of the slowest assay's LOD, plus some margin for completion of the reaction and colour change (35 min).

### LAMP reaction conditions

(i) Fluorescent detection: LAMP reactions were carried out at a final volume of 10 μL per reaction. Each mix contained the following: 1 μL of 10× custom isothermal buffer (pH 8.5–9), 0.5 μL of MgSO4 (100 mM stock), 0.56 μL of dNTPs (25 mM stock), 0.3 μL of BSA (40 mg/mL stock), 1 μL of 10× LAMP primer mix (F3/B3 2.5 μM, LF/LB 10 μM and FIP/BIP 20-40 μM), 0.25 μL of Syto9 dye (20 μM stock), 0.1 μL of NaOH (0.5 M stock), 0.04 μL of Bst 2.0 WarmStart DNA polymerase (120 K U/mL stock), 1 μL sample, and enough nuclease-free water to bring the volume to 10 μL.

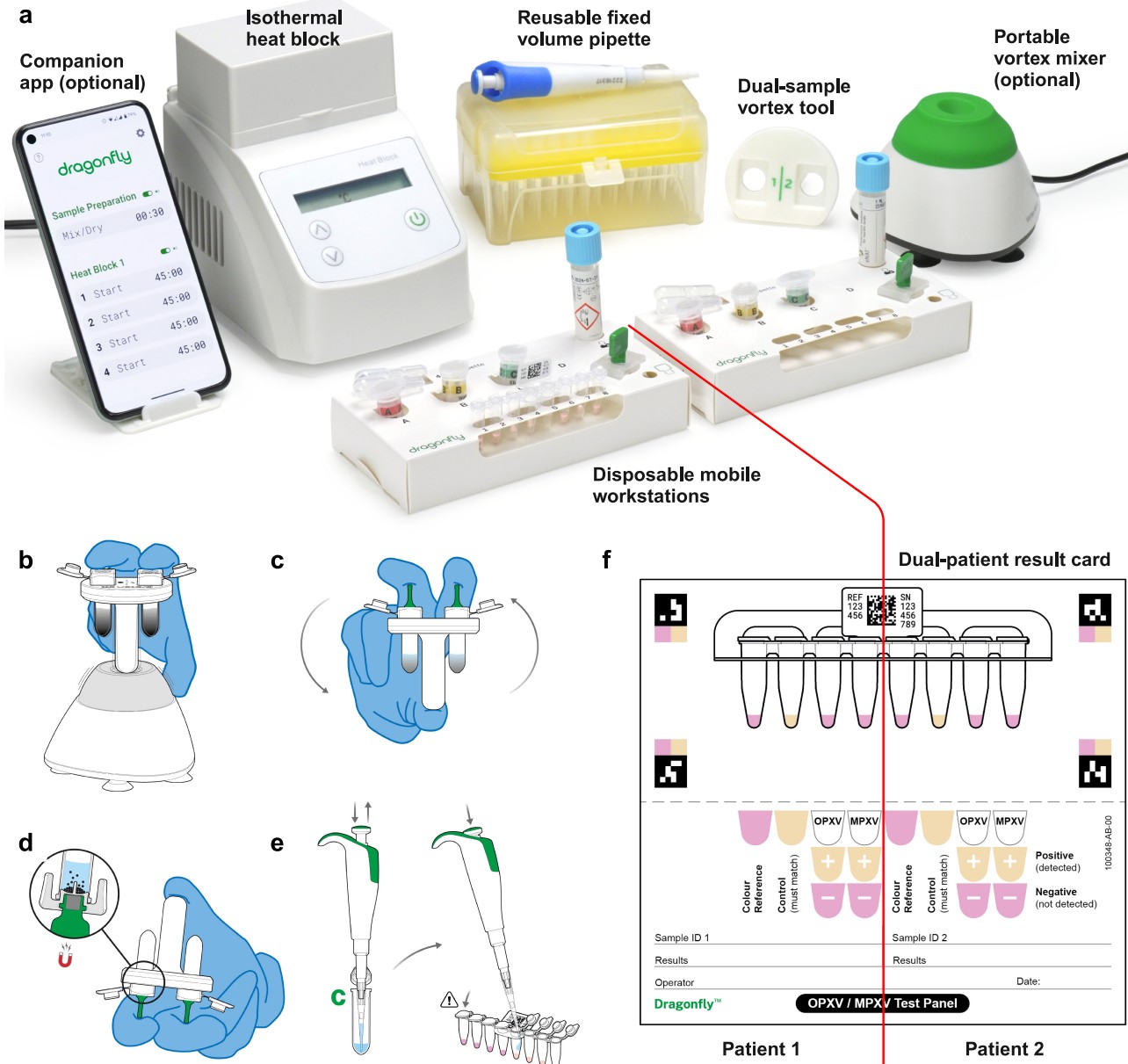

**Fig. 5 | Dragonfly OPXV/MPXV Multi-patient. a** Overview of the complete mobile testing kit, reconfigured for the simultaneous extraction and detection of OPXV and MPXV from two patient samples. **b** Dual-sample vortex tool enabling the simultaneous vortex mixing of two samples. **c, d** Magnetic bead collection performed concurrently for both samples. **e** Sequential loading of the test panel; the second half of the panel is loaded only after the tubes in the first half are sealed to minimise the risk of cross-contamination. **f** Example of a valid negative result for both patients, along with the corresponding dual-patient result card. Detailed instructions for use can be found in the Supplementary Methods. Created in BioRender. Cavuto, M. (2025) https://BioRender.com/d45n457.

Catalogue numbers are provided in Supplementary Data. Reactions were loaded into 96-well plates and were performed at 63.5 °C for 35 min using a QIAquant real-time PCR (QIAGEN) and LightCycler 96 (LC96) instrument (Roche). One melting cycle was performed at 0.1 °C/s from 63 °C up to 97 °C for validation of the specificity of the amplified products. A non-template control (NTC) was included in every experiment.

(ii) Colourimetric detection: LAMP reactions were performed using the colourimetric Dragonfly Skin Infection Viral Test Panels at a final volume of 20 μL, adding 20 μL of sample elution in nuclease-free water to each lyophilised reaction mix. Amplification reactions were carried out in a portable isothermal heat block for 35 min at 63.5 °C.

## Analytical sensitivity and specificity of LAMP assays

Synthetic DNA for each of the targets was purchased from Twist Bioscience or Integrated DNA Technologies (IDT) in lyophilised format and rehydrated in nuclease-free buffer to 5 ng/μL (sequences are included in Supplementary Table 4). Analytical sensitivity was evaluated using 10-fold serial dilutions of synthetic DNA ranging from $10^7$ to $10^1$ copies per reaction with half-dilutions between $10^3$ and $10^1$ copies per reaction (i.e., 500 and 50 copies per reaction). Dilutions were performed using nuclease-free water. Each condition was run in triplicates using an LC96 instrument (source data provided in Supplementary Data).

Analytical specificity was performed in-silico based on sequence alignments and mapping of primers, and experimentally using

extracted DNA from viral particles including AMPLIRUN® TOTAL MONKEYPOX VIRUS CONTROL (SWAB) from Vircell (reference MBTC032-R), HSV1&2 Positive Control from ZeptoMetrix (CATALOG# NATHSV-6L) and Varicella-Zoster Virus Stock (Quantitative) from ZeptoMetrix (CATALOG# NATVZV-STQ). Extractions were performed manually using the QIAamp Viral RNA Mini Kit as recommended by the manufacturer, using 140 µL as input (Fig. 4c).

## Statistical analysis

Sample size was calculated based on the equation reported by Banoo et al.[69] Considering a margin of error of 10% with a CI of 95%, and an estimated 90% sensitivity and specificity, the size of the study population had to be at least 35. TTP data is presented as mean TTP ± standard deviation.

## Extraction of nucleic acids from skin swabs and PCR reaction conditions

Nucleic acids were extracted from lesion swabs collected in Roche COBAS PCR media (P/N: 07958030190) as part of routine diagnostic service at North West London Pathology using the Qiagen Viral RNA kit for the Qiagen EZ-1 advanced Excel system, and were tested with an mpox laboratory developed test (LDT)[70]. Briefly, this consisted of a TaqMan assay targeting *G2R* WA for mpox clade II[55], and a TaqMan assay targeting *E9L* gene of OPXV[54]. The mpox LDT was reported as "mpox detected" if amplification occurred in both the *G2R* WA and *E9L*-NVAR assays. Samples were reported as "mpox indeterminate" if amplification was observed to occur in only one of the two mpox LDT qPCR assays. This procedure only informed for OPXV and/or MPXV positive.

To further evaluate the presence of other pathogens (VZV, HSV-1, HSV-2) using gold standard methods, the collected swabs were diluted in eNAT® prior to extraction, using a 1:2 ratio (sample in COBAS PCR media:eNAT®). The same samples were used for qPCR and the Dragonfly testing as detailed below. All 164 samples were extracted using the QIAsymphony® DSP Virus/Pathogen Midi Kit (Cat. No. 937055, QIAGEN) in combination with the QIAsymphony SP instrument. Briefly, the Complex 400 protocol was used, which processes 400 µL of sample as input. Elutions were performed in cooled 96 low-skirt plates with a final volume of 110 µL per sample. Elutions were tested on the day and then stored at -80 °C. Published TaqMan assays[71] as detailed in Supplementary Table 6, were used to detect the presence of VZV, HSV-1, and HSV-2. The Promega GoTaq Probe qPCR (Cat. No. A6102, Promega) was used at a final volume of 20 µL per reaction following this protocol: 2x GoTaq qPCR master mix (10 µL), 10x assay mix (2 µL), 5 µL of sample, and the remainder nuclease-free water. The final reaction concentration of primers was 400 nM for Forward and Reverse and 200 nM for the hydrolysis probe. The cycling conditions were: 1 cycle at 95 °C for 2 min, and 45 cycles at 95 °C for 3 s followed by 60 °C for 30 s. Experiments were performed using a QuantStudio5 instrument reading in FAM channel. The published human control assay from the CDC, targeting the human *RNase P* gene, was used to verify the human origin of the sample and the quality of the extraction. The Promega GoTaq Probe 1-step RT-qPCR (Cat. No. A6121, Promega) was used at a final volume of 20 µL per reaction following this protocol: 2x GoTaq qPCR master mix (10 µL), GoScript RT Mix (0.4 µL), assay ready mix (1.5 µL), 5 µL of sample, and the remainder nuclease-free water. The cycling conditions were: 1 cycle at 45 °C for 15 min, 1 cycle at 95 °C for 2 min, and 45 cycles at 95 °C for 3 s followed by 55 °C for 30 s. Experiments were performed using a QIAquant instrument reading in the FAM channel.

## Inactivation of skin swabs using Dragonfly sample preparation

The eNAT® buffer (COPAN, 608CS01M) was tested neat against VACV, or diluted 1/10 with PBS (Gibco, 11503387), against HSV-1. 1 ml of eNAT® buffer was aliquoted into screwcap tubes for the virus and cytotoxicity controls, and triplicate screwcap tubes for exposure to virus. 10 µL of cell culture media was added to the cytotoxicity control. 10 µL of virus was added to the triplicate tubes as well as the virus control tube and incubated at room temperature for the indicated length of time. After the contact time, each sample was serially diluted in cold DMEM (Gibco, 11965092) supplemented with 2% v/v FCS (Biowest, S1400-500) and 1% v/v pen/strep (Gibco, 15140122). Additionally, a neutralisation control was set up to confirm that virucidal activity was halted after the contact time by dilution. First, the cytotoxicity control was diluted 1/9 in cold DMEM and allowed to neutralise by mixing. The virus control was then diluted 1/10 in the same tube, before continuing with the serial dilution. Inactivation experiments were carried out following the protocol described in Butcher et al.[72].

Confluent monolayers of BSC40 (for VACV titration) or VERO (for HSV-1 titration) cells were used to perform the plaque assay in 6-well plates. Cell culture media was removed from the cell monolayers and 0.5 mL of diluted samples were plated in duplicates and allowed to adsorb for 1 hour at 37 °C, with rocking after 30 minutes. The inoculum was then removed, and a semi-solid overlay was added to each well (1.5% CMC (VWR, 22525.296), 1x MEM (Gibco, 11430030)) before incubation at 37 °C for 2–3 days or until plaques were countable. Titration experiments were conducted following the protocol described in Holley et al.[73].

After incubation, the overlay was removed, and the wells were washed carefully with PBS. Enough crystal violet (Sigma-Aldrich, HT90132) (1% crystal violet, 70% ethanol (Fisher Scientific, E/0650DF/C17)) was added to cover the base of each well and incubated at room temperature for at least 30 minutes before excess stain was removed and the wells rinsed with water. Plaques were counted and the titer for each sample was calculated. The experimental design for the process described above is illustrated in Supplementary Fig. 6.

## Dragonfly system Workflow

The Dragonfly workflow comprises: (i) nucleic acid extraction with the Dragonfly Sample Preparation Kits (ProtonDx Ltd, 100-104) and (ii) pathogen detection with the Dragonfly Skin Infection Viral Test Panel (ProtonDx Ltd, 100-108). First, nucleic acid extraction buffer compositions, volumes, mixing durations, and total number of steps were substantially based on the original SmartLid method for viral nucleic acid extraction[48,49]. However, due to the cited method being developed for the extraction of SARS-CoV-2 RNA from inactivating storage media (COPAN eNAT®), the protocol was further optimised to improve nucleic acid (DNA) yield from clinical isolates that were stored in non-inactivating media (Roche COBAS PCR Medium). Swabs collected in Roche COBAS PCR media and diluted at a 1:2 ratio (sample in ROCHE COBAS PCR Medium: eNAT®) were used for nucleic acid extraction with Dragonfly (400 µL input). The guanidinium thiocyanate (GTC) and magnetic bead (TurboBeads) concentrations were increased for the lysis buffer (Tube A), and two wash steps were utilised instead of one. Following this brief optimisation process, the extraction efficiency of the platform was explored and estimated, as described in Supplementary Methods. Next, volumes of 20 µL were utilised for each lyophilised colourimetric LAMP reaction to strike a balance between reagent cost, sensitivity, and ease of user liquid handling. Finally, a sufficient reaction duration of 35 minutes was determined through observing reaction colours at 5-minute intervals during the LOD experiments. This incubation duration was also confirmed to not yield non-specific amplification through negative control experiments that were incubated for up to 50 minutes at 63.5 °C.

## Platform robustness

The robustness of the Dragonfly platform was evaluated under a series of suboptimal conditions to establish viable ranges for prescribed operating parameters during analytical testing for CE-IVD self-

certification of a different panel, which was developed to simultaneously detect SARS-CoV-2, influenza A, influenza B, respiratory syncytial virus, and human rhinovirus. For each stability experiment, exploring metrics such as operating temperature range, survivability of lyophilised reagents outside of hermetic packaging, and resistance to cross-contamination between sequentially processed positive and negative samples, eNAT® medium was spiked with a 3×LOD concentration of inactivated SARS-CoV-2 viral particles. A description of each experiment, along with rationale for each considered parameter, is provided in Supplementary Methods with results summarised in Supplementary Table 7.

### Reporting summary

Further information on research design is available in the Nature Portfolio Reporting Summary linked to this article.

## Data availability

All data supporting the findings of this study are available within the article and its supplementary files. Any additional requests for information can be directed to, and will be fulfilled by, the corresponding authors. Source data are provided with this paper.

## Code availability

The source code for the Dragonfly Android application, developed for the analysis of skin infections, is publicly available at https://github.com/nmoserpdx/dragonfly-skin-infection and has been deposited at https://zenodo.org/records/14698029 [74] Additionally, we developed a Progressive Web App (PWA) for smartphones (as illustrated in Fig. 5). The PWA is designed to enhance productivity, improve time management, and streamline daily routines by offering partially customisable timers on a single screen, with seamless compatibility across devices. The source code for the PWA is accessible at https://github.com/bahp/pwa-timerhub.

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

## Acknowledgements

This work was supported by the Department of Health and Social Care (DHSC) funded Centre for Antimicrobial Optimisation (CAMO) at

Imperial College London; the Imperial College London UKRI Impact Acceleration Account [MRC – MR/X502959/1, awarded to J.R.M.]; the Wellcome Trust CAMO-Net programme [226691/Z/22/Z, awarded to A.H.]; Wellcome Trust Innovator Award [215688/Z/19/Z, awarded to P.G.]; the Biotechnology and Biological Science Research Council [BB/X011569/1, awarded to J.R.M.]; the Engineering and Physical Sciences Research Council [EP/T51780X/1, awarded to K.M.C.] and the Imperial College Research Fellowship [WDPI.G09074, awarded to K.M.C.]. In addition, this research was funded by the NIHR [NIHR134694, awarded to J.R.M.] using UK international development funding from the UK Government to support global health research. The views expressed in this publication are those of the authors and not necessarily those of the NIHR or the UK government. The authors acknowledge the support of the local diagnostic laboratory (North West London Pathology) and the National Institute for Health Research (NIHR) Biomedical Research Centre awarded to Imperial College London. A.H., S.S., P.G. and J.R.M. are affiliated with the NIHR Health Protection Research Unit (HPRU) in Healthcare Associated Infections and Antimicrobial Resistance at Imperial College London in partnership with the UK Health Security Agency (UKHSA), in collaboration with, Imperial Healthcare Partners, the University of Cambridge and the University of Warwick. The views expressed in this publication are those of the authors and not necessarily those of the NHS, the NIHR, the DHSC, or the UKHSA. A.H. is an NIHR Senior Investigator. Some graphic elements shown were created with Biorender.com (Cavuto, M. (2025) https://BioRender.com/d45n457).

## Author contributions

M.L.C. and K.M.C. contributed equally to the work. Study concept and design: M.L.C., K.M.C., M.P., and J.R.M. Acquisition, analysis, or interpretation of data: M.L.C., K.M.C., I.P., M.P., S.M., N.M., M.C., I.S., L.E., S.L., K.M.S., L.M., O.W.S., K.T.M., R.P.S., F.B., C.M.M., J.R.M. Drafting the manuscript: M.L.C., K.M.C., D.U., C.M.M. and J.R.M. Critical revision of the manuscript: M.L.C., K.M.C., I.P., M.P., S.M., N.M., M.C., I.S., L.E., SL, K.M.S., L.M., O.W.S., K.T.M., R.P.S., F.B., S.S., A.H., P.G., D.O.U., C.M.M., J.R.M. The manuscript was written through the contributions of all authors. All authors have given approval to the final version of the manuscript.

## Competing interests

The authors declare the following competing financial interest(s): M.L.C., K.M.C., I.P., S.M., N.M., M.C., K.M.S., K.T.M., P.G. and J.R.M. have or had financial interest in ProtonDx Ltd, which currently has exclusive license to the intellectual property linked to Dragonfly (WO2023131803A1), SmartLid (WO2022180376A1), and their associated trademarks. These authors declare that they do not have any other known competing financial interests or personal relationships that could have appeared to influence the work reported in this paper. The remaining authors declare no competing interests.

## Ethics

Every experiment involving clinical samples has been carried out following a protocol approved by a national research ethics committee. Specifically, this study used fully anonymised surplus samples, which was approved by the West London National Research Ethics Committee (approval no. 06/Q0406/20).

## Additional information

¹Department of Infectious Disease, Faculty of Medicine, Imperial College London, London, UK. ²ProtonDx Ltd, Translation & Innovation Hub, Imperial College London, London, UK. ³Department of Infection and Immunity, Imperial College Healthcare NHS Trust, London, UK. ⁴Department of Electrical and Electronic Engineering, Imperial College London, London, UK. ⁵Department of Microbial Sciences, School of Biosciences and Medicine, University of Surrey, Guildford, UK. ⁶Centre for Bacterial Resistance Biology, Imperial College London, London, UK. ⁷David Price Evans Infectious Diseases & Global Health Group, University of Liverpool, Liverpool, UK. ⁸The Fleming Initiative, Imperial College London and Imperial College Healthcare NHS Trust, London, UK. ⁹CBR Division, Defence Science and Technology Laboratory, Salisbury, UK. ¹⁰These authors contributed equally: Matthew L. Cavuto, Kenny Malpartida-Cardenas. ✉e-mail: j.rodriguez-manzano@imperial.ac.uk

