## [Transparent Peer Review file · Nature Communications]

Portable molecular diagnostic platform for rapid point-of-care detection of mpox and other diseases

Corresponding Author: Dr Jesus Rodriguez-Manzano

Version 0:

Reviewer comments:

Reviewer #1

(Remarks to the Author)

This manuscript developed a portable molecular diagnostic platform named Dragonfly for point-of-care detection of Mpox within 40 mins from sample to result. This is an interesting manuscript. However, there are several issues that needs to be addressed before it can be published.

1. In the introduction section, the advantages and disadvantages of the published detection methods for monkeypox should be supplemented.
2. What are the advantages of Dragonfly developed in this work over existing methods of detecting monkeypox virus? Sensitivity? Test time? Convenience? Or others. The authors need to explain this with detailed data.
3. The efficiency of nucleic acid extraction from the sample should be calculated.
4. The size of this platform should be provided.
5. Temperature, reaction time, primer pair and other factors have great influence on the LAMP process. The results of LAMP process optimization should be provided.
6. Robustness of this diagnostic platform should be tested.

Reviewer #2

(Remarks to the Author)

The manuscript "Dragonfly: A Portable Molecular Diagnostic Platform for Rapid Point-of-Care Detection of Mpox and Other Diseases" by Cavuto et al. highlights the urgent need for point-of-care diagnostics to address the global spread of Clade II mpox and sustained human-to-human transmission of the more virulent Clade I mpox. It introduces a novel point-of-care molecular diagnostic platform based on the LAMP technology called Dragonfly, which can rapidly detect the MPXV. It claims to provide a solution for monitoring mpox in low- and high-resource settings. The platform also demonstrated high sensitivity and specificity in clinical validation for potential mpox diagnosis. However, my concern is the appropriateness of the choice of this target journal. The manuscript mostly reads like a "method paper" describing the product and the steps for the assay, which is also available for their other commercial tests. I recommend that the paper would serve better in "methods" journals. My additional concerns are the claim that this assay is a point-of-care test. The number of steps and the type of manipulations are not conducive to untrained personnel. The test can be best described as "near-point-of-care."

Version 1:

Reviewer comments:

Reviewer #1

(Remarks to the Author)

The author answered all my questions. I think it's ready to be accepted.

Reviewer #2

(Remarks to the Author)

The authors have addressed my comments and concerns. I think the manuscript is ready to be accepted.

Reviewer #3

(Remarks to the Author)

The paper reports on the development of a portable molecular diagnostic device called Dragonfly, which combines power-free nucleic acid extraction with LAMP amplification of the extracted DNA/RNA and detection of the amplification results with a color change. With time-to-results being under 40 minutes, 94% sensitivity and 100% specificity validated on 164 clinical samples, the platform answers the unmet need for point-of-care detection of orthopoxviruses including monkeypox virus, and herpes simplex virus.

The paper is well-written. However, there are issues with figures in terms of small font and vagueness of figure legends. Some methods are confusing, and thus may be difficult to reproduce. For example, it is confusing how the 164 clinical swabs were processed. There are two methods for DNA extraction listed. It is not clear if the same swabs were used in both qPCR and Dragonfly assay, or there were different swabs but same patients.

The technology has been already published and in use for other viruses. The LAMP technology with colorimetric detection is not novel as well. The new knowledge is LAMP primers designed (but the sequences are not listed) and validation of the technology using a small population of clinical samples. I think Nature Communication is not a good home for such type of reports. More focused journals may be a better fit (e.g. J. Clin. Microbiology, ACS Sensors, Clin. Chem.).

Response to reviewers' comments:**Reviewer #1**

This manuscript developed a portable molecular diagnostic platform named Dragonfly for point-of-care detection of Mpox within 40 minutes from sample to result. This is an interesting manuscript. However, there are several issues that needs to be addressed before it can be published.

We appreciate the reviewer's feedback and are encouraged that Reviewer #1 has found the manuscript to be of interest for this journal.

1. In the introduction section, the advantages and disadvantages of the published detection methods for monkeypox should be supplemented.

We thank the reviewer for pointing this out, and a more detailed explanation has been provided in the main text of the manuscript, in additional to a new supplementary table (Supplementary Table 1).

Lines 67-81 (introductory section): "At present, there is no LFT available with the necessary sensitivity for mpox detection. Consequently, NAATs, such as quantitative PCR (qPCR), have remained the gold standard for mpox diagnosis due to their excellent specificity and sensitivity.³⁴ Despite these advantages, however, qPCR presents notable limitations, predominantly in POC and near-patient use.³⁵

A selection from the limited number of commercially available near-POC and true-POC molecular mpox diagnostics are listed in **Supplementary Table 1**, as identified by FIND.³⁶ These examples include PCR panels targeting MPXV, OPXV, or other pathogens (such as VZV, or HSV) that are then paired with expensive automated systems such as Cepheid GeneXpert®, QIAGEN QIAStat-Dx Analyzer, Ustar EasyNAT, or Wondfo U-Card Dx. While the majority of these platforms provide a simplified sample-to-result workflow with minimal hands-on time, the incorporation of sophisticated electro-mechanical automation leads to bulky and expensive devices, typically confining them to controlled and centralized laboratory settings. On the other hand, strictly laboratory-based assays compatible with even larger conventional instruments have further drawbacks, including the need for a cold chain and substantial manual input from skilled technicians. All these factors limit the portability and accessibility of such diagnostic tools, hindering their deployment in emergencies, at the POC, and in low resource settings.³⁷"

Supplementary Table 1. Monkeypox diagnostics landscape, as identified by FIND.

Product-Instrument (Company)	Target	Chemistry	Time-to-result	Storage conditions	Size (mm)*	Automated workflow	Reference
					Weight (kg)*		
Xpert® Mpox - Cepheid GeneXpert® Systems (Cepheid)	MPXV clade II and non-variola OPXV	Extracted, PCR	>36 minutes	2-28°C	161 x 305 x 297	Yes, hands-on time <1 min	https://cepheid.widenet/s/qmwltddz29/cepheid-xpert-mpox-datasheet-us-ivd-0998-english
					25		

QIAstat-Dx Viral Vesicular Panel - QIAstat-Dx Analyzer 1.0 (QIAGEN)	MPXV Clade I and II, VZV, HSV-1, HSV-2, HHV6 and EV	Extracted, PCR	~ 70 min	15-25°C	234 x 326 x 517	Yes	https://www.sciencedirect.com/science/article/pii/S1386653223001488
					21		
u-card dx monkeypox virus test (Wondfo Biotech)	MPXV Clade I and II	Extracted, PCR	<40 minutes	2-30°C	315 x 245 x 355	Yes	https://en.wondfo.com/pt/index116.html
					11		
EasyNAT® Monkeypox Virus Assay -EasyNat system (Ustar)	N/A	Extracted, PCR	~ 1 hour	2-8°C	390 x 300 x 470	Yes, hand-on time <5 min	https://en.bioustar.com/product/152.html
					15		
Cue Mpox Molecular Test (Cue Health)	MPXV Clade I and II	Direct, Proprietary Isothermal NAAT	~ 25 minutes	15-30°C	74 x 74 x 37	Yes	https://cuehealth.com/products/mpox-monkeypox-molecular-test
					<1		
Pluslife Monkeypox Virus Card (Pluslife)	N/A	Direct, Proprietary Isothermal NAAT	~ 35 minutes	15-30°C	101 x 91 x 65	Yes, hand-on time < 5 min	https://www.pluslife.com/companyfile/15.html
					<1		
Skin Infection Viral Test Panel – Dragonfly (ProtonDx)	OPXV, MPXV clade I and II, VZV, HSV-1, and HSV-2	Extracted LAMP	<40 minutes	Room temperature (Max range: -20-30°C)	160 x 110 x 130	No, hands-on time <5 min	This study
					<1		

*Values provided for primary reusable equipment, for example, an accompanying automated device or in the case of the Dragonfly Platform, the isothermal heat block.

Source: <https://www.who.int/news-room/events/detail/2024/08/29/default-calendar/mpox-research-and-innovation---aligning-research-response-with-outbreak-goals>

2. What are the advantages of Dragonfly developed in this work over existing methods of detecting monkeypox virus? Sensitivity? Test time? Convenience? Or others. The authors need to explain this with detailed data.

We thank the reviewer for this comment. To further clarify the advantages of Dragonfly over existing methods to detect mpox, we have added the following information:

Lines 92-99 (Introduction): “To address existing diagnostic gaps, we have developed and validated Dragonfly, a portable, molecular sample-to-result POC diagnostic platform designed for the rapid multi-pathogen detection and differentiation of skin-tropic viruses. Our platform incorporates a simple power-free nucleic acid extraction and purification method based on magnetic beads^{49,50}, which is coupled with lyophilized colorimetric LAMP technology⁵¹. This approach significantly

reduces time-to-result (under 40 minutes) and eliminates the need for cold-chain storage. It also minimizes hands-on time and the requirement for complex instrumentation. Unlike alternative diagnostic solutions, Dragonfly only requires an isothermal heat block, eliminating the need for bulky and expensive instruments, and providing a truly point-of-care format that is convenient and accessible.”

Please refer to the answer to question 1 from Reviewer #1 above for further details on the advantages compared with other available solutions. It is important to note the size and weight of competitive molecular extraction solutions, each of which also comes with a multi-thousand-pound price tag for just the base platform. In contrast, Dragonfly requires only a simple, portable, and low-cost (<£100) isothermal heat block, with minimal power consumption (<20W continuous), enabling operation via remote battery or solar panel. On the other hand, the two identified 'true-POC' alternatives to Dragonfly achieve portability and speed by omitting nucleic acid extraction, potentially reducing sensitivity, as outlined in the introduction of the manuscript. Our platform strikes a balance between high performance and true point-of-care qualities such as portability, affordability, and speed.

3. The efficiency of nucleic acid extraction from the sample should be calculated.

We thank the reviewer for the suggestion. The extraction efficiency of the Dragonfly Sample Preparation kits has now been calculated using commercially quantified mpox viral particles and digital PCR, showing an overall recovery of 37.5% (SD = 7.7%). A detailed description has been added to the supplementary materials in the new section titled "Nucleic Acid Extraction Efficiency". In addition, further data regarding the extraction performance of the SmartLid methodology can be found at <https://doi.org/10.1021/acs.analchem.4c00319>, as evaluated with a previous iteration of the protocol and buffer recipes."

4. The size of this platform should be provided.

We thank the review for noticing this omission. The dimensions and weights of the isothermal heat block (the only required piece of electrical equipment) and the consumable sample preparation tray were added to the Results section of the manuscript.

Lines 130-132: "All required buffers are pre-aliquoted in color-coded tubes (utilizing a sequential traffic-light system of red, yellow, and green) and packaged in a cardboard tray (150 x 70 x 50 mm) along with disposable exact-volume pipettes for sample input and a SmartLid for magnetic nanoparticle manipulation.”

Lines 149-152: "Combined, these three aspects provide a platform minimizes equipment requirements, requiring only a low-cost, portable, and user-friendly isothermal heat block (160 x 110 x 130 mm, <1 kg). This heat block can be powered by mains electricity, a standard 12-volt supply, batteries, or solar panels, drawing less than 20W continuously once at temperature.”

This information is now also captured in the newly added **Supplementary Table 1**, as shown above.

5. Temperature, reaction time, primer pair and other factors have great influence on the LAMP process. The results of LAMP process optimization should be provided.

We thank the reviewer for highlighting the importance of providing detailed information on the LAMP process optimisation. We agree that factors such as temperature, reaction time, primer selection, and other reaction conditions are critical to the performance of LAMP assays. In response to this comment, we have revised the "LAMP assay design" section in the Methods:

Lines 382-389 (Online Methods): "Sequences were aligned using the MUSCLE algorithm⁷² in Geneious 2023.1.2 software and several primer sets were designed and evaluated for each target gene. Primer sets were assessed based on melting

temperature (T_m), GC content and absence of secondary structures, which was verified through the built-in DNA Fold tool in Geneious. Primer candidates were tested experimentally and optimised to perform at 63.5°C, which is the operating temperature of the Dragonfly platform. The optimal primer sets were selected based on their performance in LAMP reactions, considering factors such as amplification efficiency (time-to-positive, TTP) and specificity (absence of amplification in non-template controls).”

6. Robustness of this diagnostic platform should be tested.

Thank you for the suggestion, we agree with Reviewer #1 that it is critical to include data on the robustness of the platform. Therefore, we have added the following information:

Lines 489-498 (Online Methods – new subsection):

“Platform Robustness

*The robustness of the Dragonfly platform was evaluated under a series of suboptimal conditions to establish viable ranges for prescribed operating parameters during analytical testing for CE-IVD self-certification of a different panel, which was developed to simultaneously detect SARS-CoV-2, Influenza A, Influenza B, respiratory syncytial virus, and human rhinovirus. For each stability experiment, exploring metrics such as operating temperature range, survivability of lyophilized reagents outside of hermetic packaging, and resistance to cross-contamination between sequentially processed positive and negative samples, eNAT® medium was spiked with a 3×LOD concentration of inactivated SARS-CoV-2 viral particles. A description of each experiment, along with rationale for each considered parameter, is provided in **Supplementary Methods** with results summarized in **Supplementary Table 5.**”*

Supplementary Methods & Supplementary Table 5:

“Platform Robustness

As introduced in the main manuscript, the Dragonfly platform has been preliminarily evaluated for robustness under a variety of suboptimal operating conditions. For each condition tested, eNAT® medium was spiked with a 3×LOD concentration of inactivated SARS-CoV-2 viral particles, with a “pass” requiring 3/3 positive replicates. First it was deemed important to assess the stability of nucleic acids in the elution after extraction. Accordingly, experiments were run at three different temperatures (4°C, 20°C, and 30°C) and three different time points (3, 10, and 30 minutes) per temperature. The 4°C elution tubes were stored in the 4°C cold room in our laboratories. The 20°C elution tubes were stored at room temperature. Finally, the 30°C elution tubes were stored in a GS Biotech 170L incubator at 30°C. All temperature and time points were 100% successful, enabling the claim of nucleic acid stability in elution for at least 30 minutes at 30°C. Note, while further time-points and temperatures could have been tested, it was deemed not desirable to allow users to wait longer than this from a risk and workflow perspective.

Next, due to the possibility that a user may open the foil Test Panel packaging early, exposing the strips to the outside environment, the following test was carried out to determine the maximum length of time after which the test panel could become unusable, due to likely lyophilized reagent moisture absorption and degradation. Time points of 10, 30, and 60 minutes were evaluated, after which the Respiratory Test Panels were resuspended with extracted 3×LOD samples, 30 minutes was determined to be the recommended maximum period of time after opening a test panel before it is resuspended. Similarly, it was important to anticipate a user opening the individual flip-caps of the test panel tubes prematurely, perhaps while extracting the sample, for example. Accordingly, the lyophilized reagents were exposed to air for varying time points (2, 5, and 15 minutes) at room temperature (20°C) with all tested replicates being successful. Accordingly, while longer time points could have been evaluated, it was validated that it was acceptable to leave the tube strip open, with contents still lyophilized, for at least 15 minutes. The same time points were evaluated a second time,

however this time with the test panel reagents rehydrated with elution. As expected, due to the lack of long-term room temperature stability of the LAMP reagents in liquid form, results of this experiment demonstrated a maximum recommended waiting period after test panel rehydration of up to 5 minutes. Fortunately, the included Dragonfly Heat Block is able to heat up from room-temperature to the set temperature of 63.5°C in less than that period of time, covering for the scenario that a user forgets to turn on the Heat Block prior to running a test.

Next, it was critical to evaluate the performance of Dragonfly at varying operating temperatures. To remain consistent with the recommendations ascertained from the previously described experiments, each operating temperature tested (4, 20, and 30°C) was evaluated while following the maximum waiting times for each tested “worst-case-scenario” above (i.e. waiting 15 minutes after removing test panels from their packaging, opening their tubes, waiting another 15 minutes, rehydrating the reagents, and finally waiting 5 minutes before placing the rehydrated test panels in the heat block.) Results showed a viable operating temperature range for Dragonfly of 4-30°C, as. Note, operating temperatures were achieved through the same means as the first presented stability experiment, with all components first equilibrating to that temperature, before running the entire experiment in that environment from sample-to-result.

While the Dragonfly Heat Block was developed to be firmware locked at the ideal incubation temperature of 63.5°C, a ±1°C range was also evaluated, in order to account for slight variations in the heat block calibration, or fluctuations throughout heating. All tested replicates were successful. Given that ±1°C exceeds the manufacturers stated temperature accuracy of ±0.3°C, this ensured us that heat block calibration and temperature accuracy should not affect Dragonfly test performance.

One common metric for qualitative diagnostics tests (including antigen, antibody, and molecular) is end-point stability, meaning the length of time that the result is still clearly and accurately readable for, after the test is finished. For example, lateral flow strip based diagnostic tests have notoriously short end-point stability, as capillary and evaporative effects tend to smear and blur the result indication lines. In contrast, after evaluating n=15 Dragonfly Respiratory Test Panel results for all targets for six days (stored at room temperature, or roughly 20°C), all colorimetric results were still clear and easy to read.

Finally environmental nucleic acid contamination, which can result in false-positive test results, is a common issue that faces molecular diagnostic tests due to their high sensitivity. While it is possible for this contamination to come from entirely external sources, it is also possible for a running or completed test to contaminate future results. For example, in the context of the Dragonfly system, if one of the flip-cap lids on the test panel were to open in the middle of incubation, amplicons could be released into the air that could cause false-positives in ongoing adjacent and future tests. Therefore, it is important to evaluate the likelihood of this occurring. One way to do this is by utilizing a series of alternating “high-positive” and negative samples, run in quick succession, and ensuring that all results are either true-positives or true-negatives. Accordingly, a total of 10 tests were performed on eNAT® samples, half of which were spiked with 10,000 copies/mL of SARS-CoV-2 viral particles, and the other half of which were negative. All negative samples tested were correctly identified, demonstrating a low risk of cross-contamination when performing a Dragonfly test, even when processing positive and negative samples in close succession and physical proximity.

Supplementary Table 5. Summary of preliminary robustness results.

Parameter	Result
Stability of elution	4-30°C for ≤ 30 minutes
Stability of closed test panel	≤ 30 minutes
Stability of open test panel	≤ 15 minutes

Stability of rehydrated test panel	≤ 5 minutes
Operating temperature	4-30°C
Incubation temperature	63.5°C ±1°C
End-point stability of colourimetric result	≤ 6 days
Cross-contamination	Zero demonstrated cross-contamination when alternating high-positive and negative samples (n=10)

Further work is planned to expand upon the tested ranges and tailor the examined conditions to the Skin Infection Viral Test Panel. For example, a wider operating temperature range should be tested in order to ensure applicability to the POC in warmer climates, and interfering substances likely to be found in the relevant sample type (i.e. skin lesion swabs) should be considered."

Reviewer #2

The manuscript “Dragonfly: A Portable Molecular Diagnostic Platform for Rapid Point-of-Care Detection of Mpox and Other Diseases” by Cavuto et al. highlights the urgent need for point-of-care diagnostics to address the global spread of Clade II mpox and sustained human-to-human transmission of the more virulent Clade I mpox. It introduces a novel point-of-care molecular diagnostic platform based on the LAMP technology called Dragonfly, which can rapidly detect the MPXV. It claims to provide a solution for monitoring mpox in low- and high-resource settings. The platform also demonstrated high sensitivity and specificity in clinical validation for potential mpox diagnosis.

We are also grateful for the recognition of the Dragonfly platform's potential in providing a rapid, sensitive, and specific solution for MPXV detection.

- 1. However, my concern is the appropriateness of the choice of this target journal. The manuscript mostly reads like a “method paper” describing the product and the steps for the assay, which is also available for their other commercial tests. I recommend that the paper would serve better in “methods” journals.**

We appreciate the feedback from the reviewer and would like to address the concern regarding the suitability of Nature Communications for our manuscript, as well as the perceived methodological focus of our work. While we acknowledge that our manuscript includes detailed descriptions of the Dragonfly platform and its development, we believe it offers significant advances beyond the scope of a typical methods paper. Nature Communications is a multidisciplinary journal dedicated to publishing high-quality research that represents important advances in biological, health, physical, chemical, Earth, social, mathematical, applied, and engineering sciences. Our manuscript aligns with this scope by introducing a novel platform with broad applicability and real-world impact. Specifically, the Dragonfly platform provides molecular-grade point-of-care diagnostics, with demonstrated utility in addressing public health emergencies, such as the mpox outbreak and other skin pathogen challenges, showcasing its adaptability and its potential for field deployment. This novel approach also holds the potential to extend beyond infectious diseases to fields such as cancer and other critical health domains.

We would like to highlight the following key points supporting the relevance of our manuscript to Nature Communications: (1) Novelty and Broader Implications: The Dragonfly platform introduces a fundamentally new technology that advances the field by enabling rapid and precise diagnostics at the point of care. This is not merely a methodological improvement, but a tool capable of fostering new discoveries in infectious disease surveillance and response, as well as in broader diagnostic fields, which has wide-reaching implications across healthcare. (2) Interdisciplinary Reach: Our work integrates elements from multiple disciplines, including molecular biology, diagnostic technology, and healthcare, which aligns well with the multidisciplinary scope of Nature Communications. Beyond infectious diseases, the platform can be applied in fields such as oncology and other areas requiring precise, real-time diagnostics, further expanding its relevance. (3) Potential for Future Applications: Beyond the immediate findings in the manuscript, our methodology opens doors for a wide range of future research directions. The Dragonfly platform has the potential to be adapted for new applications, both in basic and applied research, supporting innovative research that will benefit the broader scientific and medical communities.

Given these points, we respectfully request that our manuscript is considered for publication in this journal as it fits the high-impact and interdisciplinary nature of Nature Communications. We hope this clarifies the broader scientific contribution and potential impact of our work.

2. My additional concerns are the claim that this assay is a point-of-care test. The number of steps and the type of manipulations are not conducive to untrained personnel. The test can be best described as “near-point-of-care.”

We thank the reviewer for this observation and suggestion to recharacterize the test as near-POC. However, we would like to offer the following evidence to potentially retain Dragonfly’s labelling as a true-POC test in this manuscript:

1. On August 29-30, 2024, shortly after the announcement that the recent upsurge of mpox throughout Africa constitutes a public health emergency of international concern (PHEIC), the WHO hosted a scientific conference entitled: “Mpox Research and Innovation, Aligning Research Response with Outbreak Goals.” (Access a recording of the live stream at the following link: www.who.int/news-room/events/detail/2024/08/29/default-calendar/mpox-research-and-innovation---aligning-research-response-with-outbreak-goals.) There, FIND presented an overview of the current mpox diagnostic landscape, identifying only 10 molecular POC solutions, divided into 7 “near-POC” and 3 “true-POC.” Following an independent performance and usability evaluation by the WHO in Geneva, Switzerland, the Dragonfly Skin Infection Viral Test Panel and platform was featured classified as one of the only 3 “true-POC” solutions.

2. Throughout its development, Dragonfly has seen use in a variety of extreme and remote POC environments. Below are brief descriptions and images from a subset of such examples:

Example #1: 600 km north of the Arctic Circle, Infectious disease specialist and adventurer Major Scott Pallett pitched a tent and evaluated Dragonfly’s Respiratory Test Panel, which simultaneously detects SARS-CoV-2, Influenza A, Influenza B, Respiratory syncytial virus, and human rhinovirus. With nothing more than a tent and some portable solar panels and battery packs, Scott demonstrated Dragonfly’s robustness and ease of use in temperatures as low as -35 degrees Celsius and at a height of 1000 m above sea level.

Operating the platform entirely with bulky winter gloves, Scott achieved perfect results on a set of contrived specimens positive for SARS-CoV-2 and Influenza A, later recounting “Dragonfly delivers reliable results in the world’s harshest conditions. I’m confident this kit could cope with any environment.”

Example #2: Dr Jesus Rodriguez Manzano presenting the Dragonfly platform in a workshop at the June NIHR Global Health Research Group (GHRG) Conference in The Gambia.

(<https://www.digitaldiagnostics4africa.org/post/enhancing>)

Surrounding this visit, a total of 381 clinical blood specimens, stored at the MRC Unit The Gambia Biobank, and the Clinical Research Unit of Nanoro, Burkina Faso, were tested to evaluate the performance of Dragonfly's Malaria Pan/Pf Test Panel, with comparison against RDTs and microscopy. The platform met the WHO's essential criteria for community-level malaria screening, demonstrating a limit-of-detection (LOD) of 3.4 parasites/ μL , and detected 85.7% of asymptomatic cases – outperforming expert microscopy (71.4%) and rapid diagnostic tests (66.7%). The overall sensitivity and specificity for the platform achieved in this study was 95.8% [95% CI: 88.1-99.1] and 99.7%

[95% CI: 98.2-100.0] respectively. *Please note, the above data is yet unpublished, and we are aiming to submit this work as a follow-up of the manuscript under review. Please treat this as confidential.*

Example #3: The 2022 Olympic Winter Games were held at three separate locations in and around Beijing, which posed unusual challenges for monitoring the athletes' health and optimising their physical and mental well-being. Team GB brought with them three Dragonfly Respiratory Backpacks (see below), utilizing one at each olympic village.

Testing sites ranged from hotel rooms to athletic treatment centres, and even the top of the downhill ski mountain (as shown below), with Dragonfly being used to screen athletes at -20 celcius and over 2000 meters above sea level. All testing throughout the games was conducted by individuals with no prior laboratory experience or training. The image above, was adapted into new **Supplementary Fig. 5**.

- An external usability evaluation was performed in collaboration with Katalyst Laboratories, an independent United Kingdom Accreditation Service (UKAS) Medical Laboratory accredited to ISO 15189:2012. All users (n=20) were recruited by Katalyst, and the evaluation was carried out in their premises at 18 Soho Square, London W1D 3QH.

The user group was wholly selected by Katalyst and consisted of users with varying degrees of relevant experience and backgrounds. For example, user occupations ranged from trained lab technicians to non-technical staff such as receptionists, with no prior lab experience or training. As Dragonfly does not require laboratory conditions, a meeting room used for testing, which corresponded well with an example of an intended user scenario. In order to validate use with and without the software, all users were made to note their results on the card and interpret the colour change without the software, emulating offline use, in parallel to using the companion app as intended.

Training of the new users on the correct use of the Dragonfly system was provided (lasting less than one hour), after which the users performed two tests independently, one after another, told to follow instructions provided by the companion app. They returned 25 minutes later when an alarm on their tablet sounded, and removed the test panel, read the results, and recorded them in the companion software.

Throughout both rounds, users were observed for “use errors,” in which an action or lack of action resulted in a different outcome than intended by the manufacturer or expected by the user, and “close calls,” in which a use error was almost committed, but ultimately avoided. These were recorded in a form (one per user), with boxes next to each observable action (total of 60) in the Dragonfly protocol for recording the incidences and descriptions of use errors and close calls. Users were also asked to fill out a questionnaire after completing their second test. Results are summarized in the table below:

User questionnaire results				
Quality of training	Ease-of-use	Usefulness	Ease-of-learning	Satisfaction/intention to use
Ranked on a scale of 1 to 7, with 7 being the most positive				
6.8	6.4	6.5	6.6	6.5
Action reporting				
Users	Total actions	Correct actions	Use errors	Close calls
20	2400	2358 (98.3%)	23 (0.9%)	19 (0.8%)

All users successfully carried out both tests, without any observed hazardous scenarios (e.g. scenarios which could lead to an erroneous result, affecting clinical decision-making, or scenarios which could lead to user harm.) Out of a possible 2400 actions, only 23 use errors (0.9% of actions) and 19 close calls (0.8% of actions) were noted. All users filled out user questionnaires after the test. They were asked about the ease of carrying out each step in the process, quality of training, ease of use, usefulness, ease of learning, satisfaction/intention to use.

Note, this data and study above is still unpublished, and being saved for a follow-up publication. Please accept this purely as justification of use of the term “POC” in this manuscript and keep confidential.

We hope that the three primary points noted above, pertaining to the WHO’s and FIND’s external evaluation of the Dragonfly Skin Infection Viral Test Panel, examples of extreme and remote environment field studies, and a thorough external usability evaluation, should together make a case to the review that the POC label for Dragonfly is appropriate to use in this manuscript.

Response to reviewers' comments:**Reviewer #3**

The paper reports on the development of a portable molecular diagnostic device called Dragonfly, which combines power-free nucleic acid extraction with LAMP amplification of the extracted DNA/RNA and detection of the amplification results with a color change. With time-to-results being under 40 minutes, 94% sensitivity and 100% specificity validated on 164 clinical samples, the platform answers the unmet need for point-of-care detection of orthopoxviruses including monkeypox virus, and herpes simplex virus. The paper is well-written.

We appreciate the reviewer's feedback and are encouraged that Reviewer #3 has found the manuscript to be well-written.

1. However, there are issues with figures in terms of small font and vagueness of figure legends.

We have updated Figures 1, 3, 4, and 5 to enhance readability by adjusting the font size and layout. Additionally, we have expanded the figure captions to provide clearer and more detailed explanations. Furthermore, we have included a comprehensive Instructions for Use section in the Supplementary Information file (pages 19 to 34) to support understanding of the proposed methodology.

2. Some methods are confusing, and thus may be difficult to reproduce. For example, it is confusing how the 164 clinical swabs were processed. There are two methods for DNA extraction listed. It is not clear if the same swabs were used in both qPCR and Dragonfly assay, or there were different swabs but same patients.

Thank you for your comment. We apologise for any confusion regarding the processing of the 164 clinical swabs. To clarify, we have added the following text:

Line 435: "Nucleic acids were extracted from lesion swabs collected in Roche COBAS PCR media (P/N: 07958030190) as part of routine diagnostic service at North West London Pathology..."

Line 441: "This procedure only informed for OPXV and/or MPOX positive."

Line 443: "To further evaluate the presence of other pathogens (VZV, HSV-1, HSV-2) using gold standard methods, the collected swabs were diluted in eNAT[®] prior to extraction, using a 1:2 ratio (sample in COBAS PCR media:eNAT[®]), so the same samples were used for qPCR and the Dragonfly testing as detailed below. The 164 samples were also extracted using the QIASymphony[®] DSP Virus/Pathogen Midi Kit (Cat. No. 937055) in combination with the QIASymphony SP instrument."

Line 490: "The swabs collected in Roche COBAS PCR media and diluted at a 1:2 ratio (sample in ROCHE COBAS PCR Medium: eNAT[®]) were used for nucleic acid extraction with Dragonfly (400 μ L input)."

In addition, to enhance reproducibility, we have included the LAMP primer sequences (Supplementary Table 2), synthetic DNA (gBlocks) sequences used during analytical optimisation (Supplementary Table 4), and a platform robustness section in the Supporting Methods (Page 16 in the Supporting Information). Additionally, we have provided further experimental data, including other LAMP primer designs, in the Supplementary Data file.

3. The technology has been already published and in use for other viruses. The LAMP technology with colorimetric detection is not novel as well. The new knowledge is LAMP primers designed (but the sequences are not listed) and validation of the technology using a small population of clinical samples. I think Nature Communication is not a good home for such type of reports. More focused journals may be a better fit (e.g. J. Clin. Microbiology, ACS Sensors, Clin. Chem.).

Thank you for your feedback and for sharing your perspective on the manuscript. While it is true that LAMP technology with colorimetric detection has been published previously, this is not the primary focus or novelty of the work presented in this manuscript. In this study, we introduce for the first time a novel sample-to-result system (Dragonfly), which includes supporting software, novel LAMP assays in colorimetric lyophilised format (with LAMP sequences now provided in Supplementary Table 2), and an enhanced SmartLid-based sample extraction kit designed for point-of-care use and adapted for skin swabs—all of which have not been published before. Furthermore, we demonstrate the application of this technology for the differential diagnosis of MPOX, an emerging pathogen recently declared by WHO as a public health emergency of international concern.

The innovation of value of our work has recently been recognised by FIND, which has included Dragonfly as one of only three truly point-of-care molecular diagnostics for mpox. (www.who.int/news-room/events/detail/2024/08/29/default-calendar/mpox-research-and-innovation---aligning-research-response-with-outbreak-goals):

We believe the interdisciplinary nature of the work and its potential impact on global health diagnostics, particularly in under-served resource environment, make it well-suited for Nature Communications. We appreciate your consideration of our manuscript and look forward to any further comments you may have.